# Economic Education and Household Financial Outcomes during the Financial Crisis

**Paul W. Grimes** [1,2,*] , **Kevin E. Rogers** [2] **and William D. Bosshardt** [3]

1   Kelce College of Business, Pittsburg State University, Pittsburg, KS 66762, USA
2   College of Business, Mississippi State University, Mississippi State, MS 39762, USA; krogers@business.msstate.edu
3   College of Business, Florida Atlantic University, Boca Raton, FL 33431, USA; wbosshar@fau.edu
*   Correspondence: paul.grimes@pittstate.edu; Tel.: +1-(620)-235-4590

**Abstract:** Using cross-sectional data from a nation-wide survey of American head-of-households conducted in the spring of 2010, we examined the ameliorating effects of economic literacy on the probability of specific household financial outcomes resulting from the 2008 financial crisis and the associated Great Recession. A series of probit regressions were estimated to capture the impact of economic literacy on the probability that households experienced job loss, delinquent mortgage payments, delinquent credit card payments, delinquent auto loan payments, loss of home, and personal bankruptcy. The head-of-household's economic literacy was measured by the level of formal education received in economics and by the score achieved on an in-survey quiz of basic economic concepts and principles. The results indicate that realized quiz scores were correlated with the mitigation of job loss, late payment behavior, and personal bankruptcy, *ceteris paribus*. However, the results for the impact of formal economic coursework in school were mixed.

**Keywords:** economic literacy; household finances; financial crisis; Great Recession

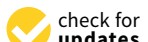



## 1. Introduction

The financial crisis of 2008 and the resulting Great Recession had a profound economic impact on a significant portion of U.S. households. While many households experienced increases in consumer debt, mortgage delinquencies and foreclosures, others experienced only minimal and transitory effects (Hurd and Rohwedder 2010; Brown et al. 2013). Investigating the sources of the disparate negative economic and financial problems, some observers have concluded that at least some of the effects resulted from poor decisions by households based on misunderstanding of their financial circumstances and operations of the marketplace (see for example, Bucher-Koenen and Ziegelmeyer 2014). We hypothesize that consumers with a documented understanding of basic economic concepts and market principles generally make more prudent financial decisions, and thus, fare better during a crisis and economic recession, than those consumers without a demonstrated degree of economic literacy. We empirically test this hypothesis by addressing the link between acquired economic education and literacy and the experiences of households during the 2008 financial crisis and the associated economic recession.

Data from a nationwide representative survey of adults were analyzed to identify and measure differential impacts of the financial crisis on households relative to the level of economic literacy demonstrated by the head-of-household. The economic literacy of survey respondents is captured by two distinct measures—(1) the extent of formal economic education received in high school and college, and, (2) the results of an in-survey quiz over basic economic concepts and principles. Probit regression equations were constructed to estimate the probability of households experiencing six separate negative financial outcomes during the crisis—job loss, delinquent mortgage payments, delinquent credit card payments, delinquent auto loan payments, loss of home, and personal bankruptcy. The

probit models are designed to control for independent variations in household demographic characteristics, geographic location, and other external variables.

The empirical results indicated that economic literacy, when measured as realized quiz scores, was correlated with the mitigation of job loss, late payment behavior, and personal bankruptcy, *ceteris paribus*, during the recession. However, the results for the impact of formal economic coursework in school were mixed. Although those with an undergraduate college course in economics were less likely to have lost a job during the recession, they were more likely to exhibit late payment behavior.

## 2. Background and Literature

Economists have a long tradition of studying the craft of teaching through the lens of their discipline (Grimes and Mixon 2021). For more than half a century, economic educators have used their classrooms as laboratories to model the production of economics human capital. The primary focus of this research continues to be on the educational production process. Economic education research is rooted in the examination of pedagogical techniques and innovation, technological enhancements, and explorations of how students learn complex economic concepts (Grimes 2019). However, the driving underlying motivation of this work is that economics is valuable, that learning the "economic way of thinking" (Jackstadt et al. 1990; Emerson and English 2016) provides a framework for making sound decisions, which in turn improves and enhances quality of life. For example, Rebeck and Walstad (2014) argue that true financial literacy and strong decision-making skills are dependent upon a solid foundation of economic understanding.

It has only been in recent years that researchers have turned their attention to examining the impact and consequences of economic education on long-term outcomes (Walstad and Rebeck 2002). For example, Allgood et al. (2004) compiled a longitudinal database of American college students to investigate the retention of economic understanding and knowledge years after leaving school. The authors analyzed these data to examine how college coursework in economics positively influenced labor market outcomes, personal financial choices, as well as enhancing long-term civic behaviors such as voting and volunteerism (Allgood et al. 2010, 2011). Likewise, Grimes et al. (2010) surveyed low-income households to examine how economic education and literacy affect the probability of owning a bank account. Their results suggest that taking an economics course in high school and demonstrating basic economic literacy are positively related to maintaining a commercial bank account. While these previous studies focused on specific groups (Allgood et al. 2010, 2011)—college students; Grimes, Rogers and Smith—low-income households), our investigation, however, examines the effect of economic education on the broader overall population of U.S. households. Furthermore, this is the first study to examine how economics understanding influenced the financial position of American households due to a major macroeconomic crisis.

While most studies of the U.S. experience with the Great Recession and the 2008 financial crisis concentrate on the underlying structural issues within the financial sector at that time (see, Financial Crisis Inquiry Commission 2011), and the policy response to the crisis (for example, Bernanke et al. 2020) those studies that examine the impact on American households are generally only descriptive in nature (for example, Hurd and Rohwedder 2010; Brown et al. 2013). These studies describe the years-long runup in consumer indebtedness prior to the crisis followed by a sharp and steady decline afterward accompanied by a rise in the savings rate.

Although much has been written about what factors allowed business firms to successfully ride out the crisis and associated recession (see Frick 2019 for a review of this literature), little is known about the underlying beneficial factors for households. Prior researchers have rarely focused on identifying the characteristics and behaviors of households which were spared the adverse consequences of the shock. However, in the spirit of our investigation, Bucher-Koenen and Ziegelmeyer (2011, 2014), using German data, reported that while financially illiterate household were less likely to own risky assets,

they were more prone to sell financial assets that lost value during the crises, thereby locking in losses. They conclude that such experiences discourage and limit future financial investments by these households.

Prior to the current study, the role of economic knowledge, and associated financial literacy, on ameliorating the negative effects of macro financial crises on household finances has been limited to a study conducted in a nation other than the United States. Klapper et al. (2013), reported that Russian householders with higher degrees of financial literacy were significantly less likely to experience negative income setbacks during the crisis and were better prepared to handle future macroeconomic shocks because they had greater access to unspent income. Our intent is to determine if economic education and revealed economic understanding mitigated the financial effects of the crisis on U.S. households.

### 3. The Survey Data

By all accounts, the financial crisis of 2008 had significant and profound effects on the U.S. and global economy. While much has been written about the causes and consequences of the financial crisis, the purpose of the present study is limited to examining if higher levels of economic literacy of household decision-makers reduced the probability of experiencing negative financial outcomes during the crisis and the associated recession. What has come to be known as the Great Recession began in December 2007 with the financial crisis unfolding throughout the fall of 2008. The recession officially ended in June 2009 (National Bureau of Economic Research 2010) followed by a long and slow economic recovery. (For an in-depth overview of the financial crisis and its impact on the economy, the reader is encouraged to visit "Visualizing the Financial Crisis" website hosted by the Hutchins Center at the Brookings Institution and Yale Program on Financial Stability (2021)).

The survey data used in this study were collected approximately one year after the declared end of the recession, during the spring of 2010. This was more than two years removed from the start of the recession and a year-and-a-half from the financial crisis. The timing of the survey allowed respondents ample opportunity to reflect on their household's experiences. The national survey was conducted by the Survey Research Laboratory of the Social Science Research Center at Mississippi State University as part of a grant program overseen by the Council on Economic Education and funded by the U.S. Department of Education. Data were collected from 1408 head-of-households across all 50 states and the District of Columbia. Detailed descriptions of the overall sample can be found in previous analyses of the survey data which focused on the respondents' opinions regarding the causes for the crisis and the policy prescriptions used to combat the recession (Evans 2013, 2015; Grimes et al. 2014). Overall, the survey sample is generally representative of U.S. households and appropriate to address the primary focus of this investigation.

### 4. Effects of the Financial Crisis

The general financial effects of the crisis on the households in our sample are summarized in Table 1. Survey respondents were asked a series of questions about their household's economic experiences "since the financial crisis." While nearly 43 percent of the sample experienced a decline in income, a little more than 52 percent had a decline in spending. The larger reduction in spending suggests households were either paying off debt or saving in preparation for expected future financial difficulties (as suggested by Bucher-Koenen and Ziegelmeyer 2014). A full 14 percent of the sample reported a job loss since the Fall of 2008, when the crisis was underway. Although this was significantly above the national unemployment rate at the time of the survey, some who reported job loss would have returned to work prior to responding to the survey. In terms of debt repayment, nearly 10 percent reported being late (more than 30 days) on housing and credit card payments with 6.5 percent being late on auto or other loans. Examining the value of outstanding mortgage debt, 16 percent were "underwater" with the value of their home below the remaining amount owed on the home. A little more than 3 percent of mortgage

holders in our sample lost their home due to circumstances resulting from the financial crisis while a bit less than that formally declared personal bankruptcy.

**Table 1.** The Financial Crisis: Effects on Surveyed Households.

| Event | Percent |
|---|---|
| Decreased Household Income | 42.9% |
| Decreased Household Spending | 52.2% |
| Lost Job | 14.1% |
| Late on Mortgage or Rent > 30 Days | 9.6% |
| Late on Credit Cards > 30 Days | 9.9% |
| Late on Auto or Other Loans > 30 Days | 6.5% |
| Mortgage "Under Water" | 16.2% |
| Lost Home | 3.1% |
| Declared Bankruptcy | 2.8% |

Table 2 provides a more detailed before-the-crisis and after-the-crisis picture of household financial experiences. The full-time employment of respondents fell from approximately 51 to 43 percent while part-time employment rose slightly from 8.7 to 9.7 percent. The unemployed and searching doubled to a full 7.0 percent, while the proportion out of the labor force grew from about 37 to 40 percent. For housing, the percentages of renters and owners remained fairly stable, surprisingly so for homeowners given the seemingly high rate of foreclosures discussed in the popular press. Turning to mortgages, coming into the crisis, nearly 86 percent of mortgages were structured as fixed rate loans with slightly more than 12 percent having adjustable interest rates. On time mortgage payments dropped from 94 to 85 percent, with increases in occasionally late, often late, and missed payments all increasing. In the case of home equity loans, only 48.4% of those with equity in their home had not used a home equity line of credit. For those losing their home, the most common reason cited was foreclosure.

**Table 2.** The Financial Crisis: Employment, Housing and Mortgages.

| Event | Pre-Crisis | Post-Crisis |
|---|---|---|
| Employment: | | |
| Full-Time | 50.9% | 43.4% |
| Part-Time | 8.8% | 9.7% |
| Unemployed and Searching | 3.3% | 7.0% |
| Out of Labor Force | 37.2% | 40.0% |
| Housing: | | |
| Renters | 16.3% | 16.4% |
| Owners | 81.5% | 81.1% |
| With Mortgage | 64.3% | |
| Type of Mortgage Held: | | |
| Fixed | 85.6% | 60.7% |
| Adjustable | 12.4% | |
| Other | 0.7% | |
| Don't Know | 1.7% | |
| Mortgage Payment Behavior: | | |
| On Time | 94.1% | 85.1% |
| Occasionally Late | 4.9% | 10.0% |
| Often Late | 0.7% | 1.9% |
| Occasionally Missed | 0.3% | 2.5% |

**Table 2.** *Cont.*

| Event | Pre-Crisis | Post-Crisis |
|---|---|---|
| Did not borrow against home equity | 48.4% | |
| Reasons for Losing Home: | | |
| Foreclosure | | 1.1% |
| Voluntarily Abandoned Mortgage | | 0.5% |
| Sold Home/Did Not Buy Again | | 0.7% |
| Bankruptcy/Involuntary Forfeited | | 0.8% |

The credit card behavior of our sample is summarized in Table 3. Comparing pre- and post-crisis numbers, payments in full and above the minimum both fell, while minimum-only payments rose from 6.5 to 9.2 percent. Failure to make a credit card payment doubled to 1.3%. The number of credit cards owned by survey respondents followed a declining pattern pre- to post-crisis. There was an increase in the proportion of households with one or no credit cards and a decrease in the number with three or more cards. Interestingly, similar proportions of households experienced increases and decreases in credit card balances and limits.

**Table 3.** The Financial Crisis: Credit Card Behavior of Surveyed Households.

| Event | Pre-Crisis | Post-Crisis |
|---|---|---|
| Payments: | | |
| Paid Monthly In-Full | 53.9% | 52.5% |
| Paid Monthly Above Minimum | 36.9% | 33.4% |
| Paid Monthly Minimum Only | 7.8% | 11.7% |
| Failed to Pay | 0.6% | 1.3% |
| Number of Credit Cards Held: | | |
| 0 | 14.6% | 19.1% |
| 1 | 16.9% | 20.6% |
| 2 | 25.5% | 25.1% |
| 3 | 15.7% | 13.9% |
| 4+ | 28.4% | 22.3% |
| Credit Card Balances Carried: | | |
| Increased | | 26.0% |
| Decreased | | 24.7% |
| Credit Card Limits: | | |
| Increased | | 18.6% |
| Decreased | | 16.5% |

## 5. Measures of Economic Literacy

Two objective measures of economic literacy were captured by the survey; the highest level of formal coursework in economics taken by the respondent, and, the respondent's score on an in-survey quiz over basic economic concepts and principles. The quiz was previously developed by the Gallop Organization to measure the economic literacy of the general population (Walstad and Larsen 1992) and consists of seven multiple choice questions on topics such as supply and demand, inflation, productivity, monetary policy, and government spending and taxation. While previous researchers have debated how to define economic and financial literacy in adults (Huston 2010), to date, the empirical research often uses the in-survey quiz approach with as few as three to five questions (Hastings et al. 2012). Table 4 provides summary statistics on our measures of economic literacy.

**Table 4.** Measures of Economic Literacy for Full Survey Sample.

| Economic Literacy | Percent |
|---|---|
| Highest Level of Economics Course Taken: | |
| None | 36.2% |
| High School | 20.7% |
| College Undergraduate | 36.5% |
| Graduate School | 6.6% |
| Correct Response to Quiz Questions Concerning: | |
| Measurement of Economic Growth (GDP) | 45.6% |
| Definition of Federal Government Deficit (Spending > Taxes) | 49.7% |
| Identify Institution Responsible for Monetary Policy (The Fed) | 43.5% |
| Example of Fiscal Policy (Taxes) | 23.9% |
| Identify Primary Determinant of Wages (Productivity) | 54.7% |
| Erosion of Purchasing Power (Inflation) | 55.8% |
| Market Determination of Prices (Supply and Demand) | 62.5% |
| Mean Score (out of 7) | 3.3 |

As seen in Table 4, more than one third of the head-of-household respondents had never taken a formal course in economics. About one-fifth of the overall sample reported that high school was the highest level that they had studied economics. Interestingly, 36.5 percent of the sample reported an undergraduate college course as their highest level and 6.6 percent had taken a graduate course in economics. These relatively high percentages of advanced study reflect the ubiquitous role that economics plays in the typical American college curriculum in which a traditional Principles of Economics course is often a component of the core general education requirements.

The results for the Gallup Quiz reveal that the survey sample scored a bit below 50 percent overall with an average of 3.3 questions answered correctly. Closer examination reveals that the proportion answering each question was also roughly 50 percent with two obvious outliers. On the low end, Table 4 reports that only about 24 percent of the respondents correctly answered the question on the role of taxes in fiscal policy, and on the high end, nearly 63 percent correctly answered the supply and demand question on how markets determine prices. Overall, the summary statistics reported in the table indicate a substantial variation in the distribution of formal economic education and revealed economic literacy across the sample. To determine how the degree of economic literacy impacted the survey respondents' household finances during the financial crisis, a regression model was built and estimated.

## 6. Probit Regression Results

To explore the relationship between economic education and the probability that a household experienced a significant negative household financial outcome during the financial crisis, several probit regression equations were estimated using our survey results. Specifically, we focused on the question, "Which of the following things have happened to you since the Fall 2008 financial crisis?" The list of outcomes provided to the respondents included whether they lost their job, were late on a mortgage or rent payment for more than 30 days, were late on a credit card payment for more than 30 days, were late on an auto or other loan payment for more than 30 days, lost their home, or declared bankruptcy. The dependent variable constructed for each of these outcomes took a 1 if yes, 0 if no.

The estimated equations included independent variables for respondents' highest level of economics course taken, score on the economic knowledge quiz, schooling level, household income level, household size, marital status, race, gender, age, and geographic location. Thus,

$$\textit{Household Financial Outcomes} = f \,(\textit{Economics Courses}, \textit{Economics Quiz Score}, \textit{Educational Attainment}, \textit{Household Income}, \textit{Household Demographics}, \textit{Location}) \tag{1}$$

*Economics Courses* is a vector of dummy variables reflecting the highest level of economics course that the survey respondent had taken—high school level, college undergraduate level, and graduate level. No economics course is the comparison reference group. *Economics Test Score* is a series of categorical variables that group performance scores on the seven-question economics quiz. The groups divide the respondents into approximately thirds; low, middle, and high, with low serving as the omitted reference category for estimation of the equation. *Educational Attainment* is measured by a series of dummy variables that indicate the highest degree obtained (those holding only an Associate's degree were combined with those with some college but without a four year degree). Those without a high school diploma or those who did not attend high school are the comparison group. *Household Income* is captured by a series of dummy variables reflecting progressively higher levels of annual household income. The specific definitions of these income ranges are reported in Table 5. The lowest level of annual income, $20,000 and below, serves as the reference category in the estimated equations. The income categories also included those who did not know their income or who refused. (In general, refused questions were coded as missing in the data since refusals on most questions were limited to a handful of observations. However, given that income was refused more often, a special category was created to maintain these observations in the investigative sample).

**Table 5.** Definition of Variables.

| Variable | Specification |
| --- | --- |
| *Household Financial Outcomes* | |
| Lost Job | Respondent experienced unemployment due to financial crisis = 1; Otherwise = 0 |
| Late Mortgage Payments | Mortgage payments 30 days or more late due to financial crisis = 1; Otherwise = 0 |
| Late Credit Card Payments | Minimum credit card payments 30 days or more late due to financial crisis = 1; Otherwise = 0 |
| Late Auto Loan Payments | Auto loan payments 30 days or more late due to financial crisis = 1; Otherwise = 0 |
| Lost Home | Respondent lost home due to financial reasons due to financial crisis = 1; Otherwise = 0 |
| Bankruptcy | Respondent declared bankruptcy due to effects of financial crisis = 1; Otherwise = 0 |
| *Economics Courses* | |
| High School | Highest economics course completed was in high school = 1; Otherwise = 0 |
| College | Highest economics course completed was in college = 1; Otherwise = 0 |
| Graduate School | Highest economics course completed was in graduate school = 1; Otherwise = 0 |
| *Economics Quiz Score* | |
| Low | Respondent's score on Gallup Quiz was 2 or less points = 1; Otherwise = 0 |
| Middle | Respondent's score on Gallup Quiz was 3 or 4 points = 1; Otherwise = 0 |
| High | Respondent's score on Gallup Quiz was 5 to 7 points = 1; Otherwise = 0 |
| *Educational Attainment* | |
| H. S. Graduate | Highest educational attainment is high school diploma = 1; Otherwise = 0 |
| Some College | Highest educational attainment is less than Bachelor's degree = 1; Otherwise = 0 |
| College Graduate | Highest educational attainment is Bachelor's degree = 1; Otherwise = 0 |
| Graduate Degree | Highest educational attainment is Master's or higher degree = 1; Otherwise = 0 |
| *Household Income* | |
| Income 1 | $20,000 or less = 1; Otherwise = 0 |
| Income 2 | $20,001 to $40,000 = 1; Otherwise = 0 |
| Income 3 | $40,001 to $60,000 = 1; Otherwise = 0 |
| Income 4 | $60,001 to $80,000 = 1; Otherwise = 0 |
| Income 5 | $80,001 to $120,000 = 1; Otherwise = 0 |
| Income 6 | Greater than $120,000 = 1; Otherwise = 0 |
| Not Known | Respondent did not know household income = 1; Otherwise = 0 |
| Refused | Respondent refused to report household income = 1; Otherwise = 0 |

**Table 5.** *Cont.*

| Variable | Specification |
| --- | --- |
| *Household Demographics* | |
| Household Size | Number of family members living in the home |
| Married | Respondent was married or cohabitating = 1; Otherwise = 0 |
| Black | Respondent self-identified as African American = 1; Otherwise = 0 |
| Male | Respondent self-identified as male = 1; Otherwise = 0 |
| Age | Respondent's age in years |
| *Location* | |
| Urban | Household living in U.S. Census defined urban area = 1; Otherwise = 0 |
| Suburban | Household living in U.S. Census defined suburban area = 1; Otherwise = 0 |
| Rural | Household living in U.S. Census defined rural area = 1; Otherwise = 0 |

Several variables were constructed to capture important aspects of *Household Demographics*. These include the number of individuals living in the household and the survey respondent's age. A set of categorical variables were also constructed for those who self-identified as married, male, and black. The telephone exchange of each respondent was used to classify the household's residence as urban, suburban, or rural, with dummy variables for suburban and rural being used to measure the impact of these locations relative to urban households. The formal definitions of each variable used to estimate our series of equations are summarized in Table 5.

The regression equations were estimated using standard probit techniques. The results are given in Table 6. Focusing first on the economic course and test score coefficients; those who had completed an economics course in college were less likely to have lost their job relative to those who had not taken a formal course in economics, *ceteris paribus*. However, no significant effect was found for those with only a high school course in economics or those with graduate training. The results also indicate that those scoring in the top third on the economics test were less likely to have become unemployed during the financial crisis, relative to low scoring respondents.

In each of the three equations with dependent variables that capture late payments, individuals that scored in the middle or high groups on the economic test were less likely to be late on payments, *ceteris paribus*. No clear pattern on late payments emerges from the estimated coefficients that measured the highest economic course completed. While those with graduate level economics were less likely to be late on mortgage/rent or credit card payments, those whose highest economics course was at the undergraduate college level were found to more likely fall behind on payments. Interestingly, respondents whose formal education in economics ended in high school were less likely to be delinquent in making late payments on automobile loans, *ceteris paribus*. Those scoring in the middle and or top third on the economics test were also found to be significantly less likely to experience late car payment issues.

The probit results suggest that losing a home (a low probability event) was more correlated with household income (unsurprisingly) than with economics education. However, a high score on the economics quiz significantly lowered the probability of losing a home, all else equal. Likewise, as expected, personal bankruptcy was negatively influenced by higher scores on the economics test.

Thus, the impact of economics human capital on household finances resulting from the financial crisis was mixed. The results provide evidence that economic literacy, as measured by test scores, may have mitigated job loss, late loan payments, and personal bankruptcy. However, the results also revealed a positive relationship between those who had taken economics at the college level and late payment behavior. Such evidence may reflect behaviors based on an over-confidence in understanding economics as demonstrated in a previous study of the survey data (Grimes et al. 2014). Overall, when the current results are considered together, it reinforces the widely held notion that *what* a person

learns from coursework and life experiences is much more important than the mere fact that a course was taken.

**Table 6.** Probit Results: Economic Literacy and the Financial Crisis' Effects on Household Finances.

| Variable | Lost Job | Late Mortgage Payments | Late Credit Card Payments | Late Auto Loan Payments | Lost Home | Bankruptcy |
|---|---|---|---|---|---|---|
| Constant | 0.304 * | −0.056 | −0.788 ** | −0.681 ** | −0.974 ** | −0.265 |
| *Economics Courses* | | | | | | |
| High School | −0.083 | −0.041 | −0.024 | −0.180 * | 0.188 * | −0.098 |
| College | −0.102 * | 0.221 * | 0.146 * | 0.481 ** | 0.133 | −0.070 |
| Graduate School | 0.091 | −0.608 * | −0.401 * | 0.207 | 0.014 | 0.524 * |
| *Economics Quiz Score* | | | | | | |
| Middle | 0.042 | −0.271 ** | −0.081 * | −0.224 * | 0.094 | −0.268 * |
| High | −0.092 * | −0.345 ** | −0.361 ** | −0.392 ** | −0.456 ** | −0.630 ** |
| *Educational Attainment* | | | | | | |
| H. S. Graduate | −0.127 | −0.045 | 0.029 | −0.090 | 0.022 | −0.018 |
| Some College | 0.069 | −0.242 * | 0.241 * | −0.386 * | 0.205 | 0.017 |
| College Graduate | −0.066 | −0.335 * | −0.024 | −0.479 * | −0.017 | −0.317 * |
| Graduate Degree | −0.162 | −0.330 * | 0.050 | −0.613 * | 0.310 * | −0.505 * |
| *Household Income* | | | | | | |
| Income 2 | −0.109 * | −0.069 | −0.073 | 0.226 * | −0.496 ** | −0.072 |
| Income 3 | −0.668 *** | −0.309 * | −0.108 | −0.079 | −0.925 *** | −0.402 * |
| Income 4 | −0.560 *** | −0.091 | −0.244 * | −0.036 | −0.795 *** | −0.016 |
| Income 5 | −0.821 *** | −0.777 *** | −0.600 *** | −0.821 *** | −0.957 *** | −0.507 * |
| Income 6 | −1.273 *** | −1.456 *** | −0.715 *** | −0.998 ** | −1.053 *** | 0.00 |
| Not Known | −0.167 | −0.406 * | 0.000 | −0.692 * | −0.662 * | −0.377 * |
| Refused | −0.789 *** | −0.274 * | −0.129 | 0.038 | −0.806 ** | −0.089 |
| *Household Demographics* | | | | | | |
| Household Size | 0.034 * | 0.144 *** | 0.123 *** | 0.103 ** | 0.034 | −0.067 * |
| Married | 0.141 * | −0.136 * | −0.148 * | 0.030 | 0.220 * | 0.325 ** |
| Black | 0.011 | 0.501 *** | 0.360 ** | 0.534 *** | 0.238 * | 0.033 |
| Male | 0.197 ** | 0.081 * | −0.156 * | −0.018 | 0.024 | −0.137 * |
| Age | −0.021 *** | −0.019 *** | −0.011 *** | −0.018 *** | −0.014 ** | −0.020 *** |
| *Location* | | | | | | |
| Suburban | 0.108 * | −0.106 * | −0.068 | 0.080 | −0.044 | −0.088 |
| Rural | −0.048 | −0.158 * | 0.335 ** | 0.321 ** | −0.047 | −0.010 |
| N | 1355 | 1355 | 1322 | 1355 | 1355 | 1195 |
| Pseudo $R^2$ | 0.109 | 0.196 | 0.125 | 0.185 | 0.118 | 0.101 |

*** $p < 0.01$; ** $p < 0.1$; * $p < 0.5$.

It is important to note that any survey cannot capture all of the idiosyncratic factors that influence household financial outcomes. Family responsibilities, health issues, unforeseen emergencies, and a myriad of other concerns can result in circumstances leading to negative financial consequences. Likewise, our empirical model cannot capture all of the systematic factors that affect the probability of adverse financial outcomes. In addition to knowledge and understanding, attitudes, opinions, and decision-making heuristics are all critical factors that can come into play. Much additional work is needed in behavioral economics to fully understand how intrinsic feelings toward money management influence financial outcomes—and how those feelings may be manipulated (Akerlof and Shiller 2015). Regardless of these caveats, the current results suggest that revealed economic understanding may mitigate the probability of negative financial outcomes during a major macroeconomic downturn.

## 7. Conclusions

The financial crisis of 2008 was a global event that permeated the collective conscious of households during the intervening years. Not only did the financial crisis quickly become a standard topic found in economics textbooks (Register and Grimes 2016) and change the way central bankers conduct monetary policy (Ihrig and Wolla 2020), many popular press books (e.g., Blinder 2013) and even an award-winning commercial film (McKay 2015)

have explored the social issues it wrought. Unfortunately, policymakers will never control the causal factors of all fluctuations and disruptions in the global economy. Recent events have clearly shown that major financial crises can be caused by forces originating outside the marketplace. In light of the world-wide COVID-19 recession that began in 2020, it is important to understand how households with varying degrees of economic knowledge are prepared to handle a major crisis.

Our results indicate that demonstrated economic literacy, as revealed by quiz performance, was beneficially correlated with the mitigation of job loss, late payment behavior, and declaration of personal bankruptcy, *ceteris paribus*. This benefit was significant, even after accounting for overall educational attainment and previous economics coursework. However, our findings concerning the direct impact of formal economic coursework in school were mixed. Although those with an undergraduate college course in economics were less likely to have lost a job during the recession, they were more likely to exhibit late payment behavior. Additional research is needed to identify the source of this last result—is it an outcome of poor decisions due to overconfidence in understanding the changing economic environment, or is it rational behavior based on the relatively higher future income earning capacity of those who attend college? As the world recovers from the latest economic recession, the opportunity to expand this line of research will present itself. However, the results presented here suggest that households headed by those with revealed economic literacy are less likely to experience negative impacts during a major macroeconomic downturn as measured by important financial outcomes. This is an important conclusion for policymakers concerned with mitigating the effects of future financial crises and economic recessions. Investments in enhancing access and strengthening delivery of fundamental economics education have the potential to reduce the negative consequences of negative economic shocks.

**Author Contributions:** Conceptualization, P.W.G., K.E.R. and W.D.B.; methodology, P.W.G., K.E.R. and W.D.B.; formal analysis, P.W.G., K.E.R. and W.D.B.; data curation, P.W.G., K.E.R. and W.D.B.; writing—original draft preparation, P.W.G., K.E.R. and W.D.B.; writing—review and editing, P.W.G., K.E.R. and W.D.B.; project administration, P.W.G., K.E.R. and W.D.B.; funding acquisition, P.W.G., K.E.R. and W.D.B. All authors have read and agreed to the published version of the manuscript.

**Funding:** Financial support for this project provided by an Excellence in Economic Education subgrant from the Council for Economic Education through funding from the United States Department of Education Office of Innovation and Improvement. Sponsor number: GC-0901668.

**Institutional Review Board Statement:** The study was conducted according to the guidelines of the United States' *Federal Policy for the Protection of Human Subjects*, and approved by the Institutional Review Board of Mississippi State University (MSU IRB Study #09-287 for project #SPA 361075-042000-021000; approved February 2010).

**Informed Consent Statement:** Informed consent was obtained from all subjects involved in the study.

**Data Availability Statement:** Permission to review the survey data reported in this paper may be requested by contacting the authors.

**Acknowledgments:** An earlier version of this paper was presented at the Allied Social Science Associations conference. Thanks to the reviewers of this journal who provided insightful and valuable comments that improved our work. Special appreciation is extended to Marybeth Grimes for editorial assistance.

**Conflicts of Interest:** The authors declare no conflict of interest. The funding sponsors had no role in the design of the study; in the collection, analyses, or interpretation of data; in the writing of the manuscript, and in the decision to publish the results.

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
