# Peer review of "Economic Education and Household Financial Outcomes during the Financial Crisis"

_jrfm, doi:10.3390/jrfm14070316_

Round 1
Reviewer 1 Report
The paper is interesting, it deals with an important topic of the relationship between economic education and household financial outcomes during the financial crisis, and it is my pleasure to review it.
The paper has merits, is concise, organized, and uses a solid logical tool. Methodology and approaches are interesting and systematic.
However, I would have some considerations and suggestions for improving the quality of the article.
The manuscript does not develop literature review, necessary in understanding the value and usefulness of the topic. Some interesting contributions are mentioned Conclusions (!) but transitory and underdeveloped. In other words, there is no dedicated section highlighting the most important contributions in this field - the role and importance of economic and financial literacy, the relationship between economic literacy and crises, similar research (if any) on this topic, and how current research is integrated in the international flow of research on this topic. The authors approach the effects of crises in an interesting and well-structured way, then move quickly to MEASURES OF ECONOMIC LITERACY, but this section is more about methodology and date sources.
The authors used the results of survey (Spring 2010) to identify the relationship between economic literacy and several specific household financial behaviours, resulting from the 2008 financial crisis and the following recession (especially the impact of job loss, late loan repayments, personal bankruptcy, etc.).
I would therefore recommend the authors to revise the text as a result of the following suggestions:
- if there is not too short the time interval between the outbreak of the crisis, its successive impact on households and the changing behaviours (eg the tendency to delay the loans re-payments), and, respectively, the awareness of a specific behavioural pattern to be caught in a survey . For example, the delay in credit payments (and especially the systematic recurrence of these delays) generates an awareness of delinquent behaviour with a certain delay in time.
- What is now the relevance of analysing data collected over 10 years ago, highly emotional impacted by the crisis’ effects on individuals and households? How do the authors intend to draw general useful conclusions, beyond the special context of the 2008-2009 crisis? What measures could be taken to mitigate the adverse effects on households, in the event of similar or comparable crises - for example, the economic implications of the current pandemic crisis, or others.
Thank you for the opportunity to review this article and good luck!
Author Response
Thank you for your helpful and constructive comments. Our response to each can be found in the attached file. Please see the attachment.

Reviewer 2 Report
The article deals with the important problem of the connection between economic formal and real knowledge, and households coping with economic and financial shocks caused by external factors (e.g. macroeconomic). The discussed topic is worth publishing in the journal.
My critical remarks relate to the following:
1) The presented results relate to the relatively distant time of the financial crisis and the Great Recession. A dozen or so years have passed and many articles were probably published on the issues of factors determining the resilience of households to economic and financial shocks. Why are the authors not referring to other studies?
2) The results of statistical analyzes are not unequivocal. Generally they confirm the importance of economic knowledge for avoiding financial troubles and maintaining financial security. In my opinion, however, the questions in the economic quiz are too simple and there are too few of them to fully reliably assess the economic competence of the head of the household. The authors do not refer to this weakness of the research.
3) There is no reference to whether the problems of some households result from low financial knowledge and the resulting erroneous decisions, or from specific economic attitudes determined by attitudes towards risk and cognitive and decision-making heuristics.
4) The authors argue that in the United States, economic education at various levels of education is of great importance. Therefore, the question arises: why in the United States so easily and on a large scale, financial institutions "caught suckers" (reference to the book: G.A. Akerlof, R.J. Shiller, 2015: Phishing for Phools. The Economics of Manipulation and Deception)?
Author Response

(The authors gave the same response as above.)

Reviewer 3 Report
In the present paper, the authors aim at measuring the impact of household’s “economic literacy” on their fate during the subprime crisis. Economic literacy is measured in two ways. First, authors have data on agents/household school attainment and in particular on their last economic courses. We can call the “academic economic literacy”. Second, they have data from a survey indicating for each household of their “informal economic literacy”. Authors perform probit regressions over six binary variables, (Lost Job, Late Mortgage Payments, Late Credit Card Payments, Late Auto Loan Payments, Lost Home and Bankruptcy) over academic economic literacy, informal economic literacy, income, school attainment, and some demographic variable. They found that if “academic economic literacy” has merely no effect, the “informal economic literacy” has a positive effect (i.e. reduce the occurrence of the events describe by the six variables taken as an outcome of the crisis).
Two comments:
-The absence of any significant effect of the variables describing the “academic economic literacy” may come from some “correlation” with “Educational Attainment”.
-P. 20 the levels should be 0.05, 0.01 and 0.001 ?
Author Response
Thank you for your comments. Your review was appended to our submission portal after we had completed a revised manuscript based on the comments provided by two other referees. Attached you will find our detailed responses to their extensive comments.
Regarding your two points:
- Yes, the two measures of economic literacy are naturally correlated . . . but checks reveal that it is not the extent as might be expected. Previous research suggests that real life appears to trump formal education in terms of learned financial behavior. Our results are consistent with this. Regardless of the existence of multicollinearity between these vectors, the relationship does not invalidate the results. within the framework of the model.
- The prob values are correctly reported.

Round 2
Reviewer 1 Report
The authors have properly responded to our suggestions, in a very detailed manner, both by defining a literature review section and by explanations and suggestive comments, well-constructed, to some specific issues raised during the review.
In this form the paper is suitable for publication.